# Gender Differences in Sports News Coverage on Twitter

**DOI:** 10.3390/ijerph17145199

**Published:** 2020-07-18

**Authors:** Clara Sainz-de-Baranda, Alba Adá-Lameiras, Marian Blanco-Ruiz

**Affiliations:** 1Department of Communication Studies, Faculty of Humanities, Communication and Library and Science, University Carlos III of Madrid, Getafe, 28903 Madrid, Spain; cbaranda@hum.uc3m.es; 2University Institute on Gender Studies, University Carlos III of Madrid, 28903 Getafe, Spain; mangeles.blanco@urjc.es; 3Department of Business Administration, Faculty of Law and Social Sciences, University Carlos III of Madrid, 28903 Getafe, Spain; 4Department of Communication Sciences and Sociology, Faculty of Communication Sciences, University Rey Juan Carlos, 28943 Fuenlabrada, Spain

**Keywords:** sport, gender, twitter, mass media, journalism, female athletes, women

## Abstract

Gender stereotypes influence boys’ and girls’ self-perception, with the differential treatment received by sports figures in the media being one of the main factors in the perpetuation of stereotypes about sports. The objective of this research is to analyze if the new communication channels, such as Twitter, maintain gender stereotypes when reporting sports news. For this purpose, the four most followed media in Spain were analyzed: (*@ElPais_Deportes*, *@ABC_Deportes*, *@Marca* and *@MundoDeportivo*) over a period of five months, from March to June 2016. Our sample was composed of 6544 tweets, with 96.19% about sportsmen compared to 3.81% that portrayed women. The sport with the most media coverage was football (72.11%), for men as well as for women, followed by basketball (6.63%). It is clear that despite the growing international triumphs of Spanish women athletes in recent years, the latter continue to be underrepresented in the media. Female athletes receive more media coverage according to the sport which they engage in (“gender-appropriate” sports), with the exception of football, and not in accordance with their accomplishments. Twitter remains at the service of traditional media replicating the same gender biases and even augmenting them.

## 1. Introduction

Modern sport is a social event and sociocultural product that has evolved, beginning as a sign of status among privileged sectors in 19th century English society up to its current professional and amateur mass-scale practice, fully participating in all of the transformations that accompany social modernization processes [1,2]. Communication media has played a very important role in this process and its analysis renders an account of the predominant status quo during each moment in history.

There is a clear link between the practice of sport for girls and their later interest in the sport as adults; a high degree of participation in sport during school years translates into a 76% likelihood of having a lasting interest in the sport [3]. Gender stereotypes influence a self-perception that boys and girls have as of the age of six [4], with the differential treatment in the media received by women athletes influencing in perpetuation of stereotypes about sport. The main studies carried out have focused on the traditional communication media (press, radio and television) confirm that these media have reinforced gender stereotypes by reporting on male and female sport in a different way [5,6,7,8].

Today the media focus is on Internet, with the International Telecommunication Union (ITU) estimating that 53.6 per cent of the global population, or 4.1 billion people, are using the Internet [9], a percentage that has risen considerably in highly connected countries like Spain, where more than 80% of the population is connected to the Internet [10].

Within the digital media ecosystem, social networks have acquired great social relevance [11], joining traditional socializing agents (family, school, peer groups and communication media) in information processes and identity construction [12,13]. Twitter has become one of the most used social networks as a news media by stakeholders [14,15] as well as by journalists [16,17], and of course by its users [18], being employed by very diverse audiences to get information on news events [19,20,21,22].

One of Twitter’s main features is that it allows an unlimited number of tweets to be sent [23]; that is, as long as the messages have no more 280 characters there is no space or time constraints. This is an excuse that has been made in the past to justify the fact that women’s sports have less media coverage in traditional media or [24] why media reports more on sports considered “appropriate” for women [25], contributing to perpetration of gender roles. A study by Hull [26] shows that, despite the opportunities provided by Twitter to send as many tweets as desired, it is used as an extension of traditional media and not as an opportunity to share additional information for which there is no room in traditional media. Only 1.6% of sports news on Twitter has to do with sports by women [27].

The aim of this research is to analyze, from a gender perspective, media coverage by sports news media in their Twitter accounts. Our research question is: Do the new communication channels, such as Twitter, maintain gender stereotypes when reporting on sports? It is based on the hypothesis that, besides the fact that the Internet has eliminated the physical space constraint, sports news media—specialist as well as general news outlets—continue to foment the perception that there are sports “for” women and sports “for” men. These gender stereotypes end up influencing girls and young women taking part in sports.

## 2. Materials and Methods

A content analysis methodology was used for each tweet published on the days and in the media selected, examining the type of sport engaged in by the men as well as women who receive media coverage.

### 2.1. Instrument

For the content analysis, an analysis sheet with six items was used: date, communication media, sport, photo, gender represented, and in the tweets about a woman, whether the analysis differentiates between two profiles for women: athletes and non-athletes (relatives of male athletes, well-known women and celebrities, etc.) [28].

In the item “Sport,” which gathers the sports referred to in tweets from the sports media Twitter accounts, the classification by Spanish Olympic Committee (COE) has been followed, with a list of 67 sports. Nine that were not on the list were added and a variable “Olympic Games” for general news about preparation for sports was also added. Furthermore, a section of “none” was added for those tweets that appeared in sports news accounts but had no reference to any sport.

### 2.2. Procedure

In the first place, Spanish communication media with mass Internet dissemination were selected. For this purpose, the Estudio General de Medios de España (General Study on Spanish Media) (EGM, 2015) was taken as reference for the number of followers in their digital channels, specifically that of Twitter. The final selection was composed of general news media with a sports section, @ElPais_Deportes and @ABC_Deportes, and two specific sports news media, @Marca and @MundoDeportivo.

Second, through a simple random sampling [29] a day of the week was chosen to analyze the published tweets, with 1 March 2016 as the first day analyzed and 26 June 2016 as the last day. Throughout these five months the following days were analyzed: 1, 9, 17 and 25 March; 2, 10, 11, 19, 27 April; 5, 13, 21, 29 and 30 May; 7, 15, 23 June and 7, 9, 17, 18 and 26 July, for a total of 22 days. All of the tweets for the 24 h of each day were collected to avoid any skewing from the different time slots given the lack of defined guidelines for tweet publication [30]. The procedure for collecting the tweets was done manually. In addition, only the tweets posted in the mentioned accounts were counted, but not retweets (RT).

It should be noted that we decided to conduct an analysis on the media coverage of sports media during the routine period, excluding the Olympic Games, because previous studies [5,31,32] show that during the Olympic Games (an extraordinary period of coverage), female athletes receive more and different media coverage than during any other sporting event. Therefore, it was necessary to analyze what such coverage is like during a routine period.

### 2.3. Sample

The sample of the study was composed of 6544 tweets published during a five-month period from March to July 2016 in the Twitter accounts of the selected sport communication media: for *@Marca:* 2242 tweets were published, for *@MundoDeportivo*: 3121 tweets, for *@ElPaís_Deportes*: 541 tweets and for *@ABC_Deportes:* 640 tweets.

### 2.4. Data Analysis

Statistical analysis was carried out with SPSS (IBM SPSS Statistics, Armonk, NY, USA, version 21.0; EE.UU.). Contingency tables for the bivariate and descriptive analysis were used based on the observed results, aimed at specifying and analyzing the characteristics of the sampling used in the study. A Cramer’s *V* coefficient test was done by data type and number of observations to find correlations between variables and quantify the strength of that relationship. The values were estimated using Cramer’s *V* with the following criteria: >0.3 (moderate) and >0.6 (strong). Additionally, non-parametric inferential techniques were carried out (hypothesis test: $\chi^{2}$ Test of Independence) to study the possible significant relationships between the analyzed variables. The significance level was set at *p* < 0.05.

## 3. Results

In Table 1 the data reveal how the possibilities offered by Twitter for communication, without time or space constraints, do not translate into a greater presence of women athletes. In global terms we can observe that out of the total of tweets published by the analyzed media (n = 6544), men were featured 96.19% of the time with respect to women, who were the subject of the tweets 3.81% of the time.

The media that published proportionally more tweets on women was *@ABC_Deportes* (4.84%), with *@Marca* (2.01%) publishing the lowest number, which points to scant representation by women in news coverage in Twitter sports media.

Through the $\chi^{2}$ Test of Independence, a significant relationship was found between the variables “gender represented” and “media” (*p* = 0.000), which implies that the media influences the gender represented. Additionally, due to the value provided by the Cramer’s technique (*V* = 0.068) that quantifies the intensity of the relationships, it can be confirmed that this relationship, despite being significant, was not strong (Table 2 and Table 3).

In Table 4, “accompanying photo by media” and “gender represented” are analyzed. In general, 80.74% of the tweets were accompanied by an image to illustrate the information, something usual in this social network to provide more impact. Nevertheless, it was *@ABC_Deportes* (95.31%) and *@Marca* (93.84%) who used this feature most, followed by *@MundoDeportivo* (71.84%) and *@ElPaís_Deportes* (60.48%).

In the case of the variables “media” and “photo,” through the $\chi^{2}$ Test of Independence a significant relationships was found between the variables (*p* = 0.000). Additionally, due to the value provided by Cramer’s technique (*V* = 0.312), we can affirm that this relationship was moderate.

If we observe the data in Table 4 according to gender represented, it is clear that the frequency of use of photos in the tweets featuring women was very high in all of the accounts analyzed, reaching 100% of the tweets where women were represented in the account of *@ABC_Deportes*. It is evident that the use of the female image as a lure was a constant in the Twitter accounts.

Through the $\chi^{2}$ Test of Independence, a significant relationship was found between the variables “gender represented” and “photo” (*p* = 0.000), which implies that photos influence the gender represented. Additionally, due to the proportional value of Cramer’s technique (*V* = 0.069), it can be confirmed that this relationship, despite being significant, was not strong (Table 2 and Table 3).

Taking as an analysis variable the sport that the tweets dealt with (Table 5), a very high percentage of information was found for football featured as the main sport (72.11%), followed by basketball (6.63%), motor racing (3.3%), tennis (3.06%), cycling (3.01%), motorcycling (3%), Olympic movement (0.76%), boxing (0.76%), athletics (0.69%), golf (0.64%), handball (0.52%) and swimming (0.21%). In 3.4% of the tweets there was no reference to any sport.

The variable “others” (2.26%) grouped various sports together which were covered in the sports news sources, but which did not represent more than 0.2% of the overall calculation (indoor football, ice skating, rugby, “American” football, sailing, hockey, taekwondo, beach soccer, paddle tennis, archery, equestrianism, water polo, gymnastics, chess, canoeing, mountain climbing, weightlifting, snowboarding, roller skating, skiing, volleyball, judo, badminton, fencing, triathlon, and others).

Of the data obtained, it can be clearly seen that football predominated for men as well as women, with a total of the news featuring women representing 20.88% for football. Inferential techniques cannot be applied due to the small sample sizes of some categories of the variable sports.

If the data is analyzed according to the variable gender, an underrepresentation of women can be verified at the sample level in the majority of the different sports, especially if contrasted with the data of the sport engaged in. By sport, women were represented more in swimming, where 92.86% of the tweets were about women, and mainly for the swimmer Mireia Belmonte. The other sports were mainly represented by men, and tweets about women only surpassed 15% for athletics (15.56%), Olympic sports (16%), tennis (22%) and other sports (22.97%), but with percentages very far from those of their male counterparts.

Worthy of note was the elevated presence of women in tweets that made no reference at all to sport (33.17%), with this being the second most common type of news in which a woman was most likely to appear.

Through the $\chi^{2}$ Test of Independence, a significant relationship was found between the variables “gender represented” and “sport” (*p* = 0.000), which implies that the sport influences the gender represented. Additionally, due to the value obtained by Cramer’s technique (*V* = 0.403), it can be confirmed that this relationship was moderate (Table 2 and Table 3).

### Women in Sports Tweets

In Table 6, the distribution of the sample profiles of women represented are analyzed (athletes vs. non-athletes) of the tweets according to the communication media. It can be observed that 59.84% of tweets featuring women were published by the account of *@MundoDeportivo*, 18.07% by *@Marca*, 12.45% by *@ABC_Deportes* and 9.64% by *@ElPaís_Deportes*.

When differentiating between women profiled as athletes or non-athletes, it is worthy of note that tweets with female non-athletes as the subject represented 44.58%. In this case, there were very large differences between the different accounts: in *@ElPaís_Deportes* there were no tweets about female non-athletes published, in *@MundoDeportivo* these tweets represented 57.05%. These data show the very limited representation of sports women in the sports media news coverage on Twitter.

Through the $\chi^{2}$ Test of Independence, a significant relationship was found between the variables “women’s profile” and “media” (*p* = 0.000), which implies that the media influences the publication of the woman’s profile. Additionally, due to the value obtained by Cramer’s technique (V = 0.384), it can be confirmed that this relationship was moderate (Table 7 and Table 8).

The differentiation in the analysis of the women’s profiles exacerbates the invisibility of female athletes.

In Table 9, in general it can be observed that 94.42% of tweets about woman were accompanied by an image, although this was not the same in all of the media; in all the news with women as the subject published in *@ABC_Deportes* there was an accompanying picture, while in *@ElPaís_Deportes* this was true in only 75% of the cases.

Through the $\chi^{2}$ Test of Independence, a significant relationship was found between the variables “media” and “photo” (*p* = 0.000), which implies that the media influences when to publish the tweets with an accompanying image. Additionally, due to the value obtained by Cramer’s technique (V = 0.284), it can be confirmed that this relationship, despite being significant, was not strong (Table 7 and Table 8).

When analyzing accompanying image differentiating the women’s profile, we can observe that tweets featuring female non-athletes were most frequent, and in the case of *@Marca* and *@ABC_Deportes,* they reached 100%.

Through the $\chi^{2}$ Test of Independence, a significant relation was found between the variables “women’s profile” and “photo” (*p* = 0.019), which implies that the profile of the female athlete influences publishing the tweets with an accompanying picture. In addition, due to the value obtained by Cramer’s technique (V = 0.149), it can be confirmed that this relationship, despite being significant, was not strong (Table 7 and Table 8).

In Table 10, the distribution of the sample profiles of the female subjects are analyzed (athletes or non-athletes) according to the sport. The female athletes appeared chiefly in information about tennis (30.43%), football (12.32%) and basketball (10.14%). Other sports for which Spain has world class figures—such as badminton, weightlifting, canoeing and water polo—represented no more than 0.72%, and in others such as karate or handball there was no reporting.

What is most striking is that a large part of the tweets about women did not refer to any sport (none = 26.69%), and in sports such as motorcycling (66.67%), motor racing (100%) and football (67.31%), women appeared mainly as non-athletes.

Female athletes appeared most frequently in the general press sports accounts; for *@ElPaís_Deportes,* all of them were athletes. It was in the accounts of the sports papers, *@Marca* (44.44%) and *@MundoDeportivo* (57.62%), where, despite the fact that they published more tweets where women were represented, a great deal of relevance was given to the non-athlete profiles common in football or with other news or information that had nothing to do with sports.

Through the $\chi^{2}$ Test of Independence, a significant relationship was found between the variables “women’s profile” and “sport” (*p* = 0.000), which implies that sport influences the publication of the woman’s profile. Additionally, due to the value obtained by Cramer’s technique (V = 0.802), it can be confirmed that this relationship was strong (Table 7 and Table 8).

## 4. Discussion

The results obtained after analyzing the gender differences in sports news coverage in Twitter for the Spanish sports media (*@ElPais_Deportes*, *@ABC_Deportes*, *@Marca* and *@MundoDeportivo*) show that the sport with most media coverage during the five months analyzed, March to July 2016, were football (72.11%), followed by basketball (6.63%).

Football is the sport with most news for males as well as females, and it is the main sport in the four analyzed media. Of the total tweets with women as the subject, football represents 20.88%, while 74.14% are about men. We should keep in mind that football, in Spain as in other countries, is the sport with the most fans and is also the most played sport [33], in addition to being a business that involves hundreds of millions of Euros.

As for women’s football, the data show a change in the trend with respect to data obtained from the traditional press historically, where female athletes received more limited coverage in team sports [7,28,34,35,36], and in the case of football, it was in anecdotal news items.

This trend began in 2015 with the creation of the Women’s Football Club Association with the aim of improving the visibility of women’s football on television and social networks [37]. This association had its positive effects as the data in this study shows. Furthermore, in August 2016, the energy company Iberdrola announced that it was becoming the main sponsor of the First Division Women’s League of Spanish football. The injection of capital through the sponsor had an impact on the visibility of women’s football with investment in the news media for its coverage, and later, for broadcasting the games on public and private channels [38,39]. This led to increasing news coverage significantly, with football becoming consolidated in the traditional media [40]. This data is reaffirmed in our study, which points to football as the sport with female athletes receiving most coverage on Twitter.

Basketball is the sport with the second highest coverage (6.63%) and this trend is also confirmed in studies on traditional press [36,41,42,43]. Of the total of tweets on basketball, those about women represent 3.23% while those about men represent 96.77%; such differences are quite striking if we take into account that Spain’s national women’s basketball team has international triumphs among its achievements, similar to the men’s national team. In 2014, the Spanish women’s basketball team won a silver medal at the World Championship and would be silver medalists again at the Rio 2016 Olympic Games, a better position than the men’s team. This inequality is perpetuated in Twitter as had already occurred in the print media [44]. In addition, if we examine the practice of sport in Spain, this sport is the leader in female licenses, with 112,266 women becoming federated in the 2016 season, 32.35% of the total number of federated members.

In the rest of the tweets, the data confirms the underrepresentation of women in the majority of the different sports, especially if we contrast the data of the international achievements of female athletes and women’s teams. Coverage of the women depends on the sport they engage in and not on the accomplishments in the sport; in sports such as badminton, karate, weight lifting, cannoning, handball and water polo, Spanish female athletes who are world-class sports figures are made invisible [45,46]. It is the communication media themselves that offer a masculine and elitist vision focused on a reduced number of sports [40,43], where only on a few occasions do they report on other socially accepted feminine sports that are not, for example, swimming or athletics [47]. According to the data obtained, women stand out as the subject of the tweets in comparison to men only in those on swimming, where 92.86% are about women (*V* = 0.403).

Not all the media behave the same in the Twitter accounts. The relationships between the variables “gender represented” and “media” and “women’s profile” and “media” were confirmed, but this does not mean that any of them are exemplary in this sense. For example, *@ElPaís_Deportes*, although it does not publish any information on women non-athletes, barely reports at all on sports women, and as such does not help to create female athlete role models; it is purely male-dominated news and information. Another example is the opposite case, *@MundoDeportivo*, which is the account that publishes most on women, but it also is the media that publishes most about female athletes and non-athletes, fomenting a stereotyped vision of women.

Focusing on analysis of tweets featuring women, female athletes appear, above all, in football (20.88%), followed by tennis (17.67%), basketball (5.62%), swimming (5.22%) and athletics (2.81%). These results corroborate the tendency found in these previous studies which highlight that tennis, basketball [27,44] and athletics are the sports with the largest coverage within women’s sports [28,48,49]. Although there is no justification for gender of a male or female athlete to “affect the quality or quantity of coverage they receive” [50], the data reveals just the opposite.

Women who are not athletes (family members of male athletes, well-known women and celebrities, etc.) who are the subject of news and information represent 45.02% of the tweets that women are the subject of. This fact has repercussions on the stereotyping of women and renders female athlete role models invisible. These women are, fundamentally, the subject of news or information that is not about any sport (58.41%) or related to football and football players (30.97%), a practice previously verified in studies carried out on print media in Spain between 1979–2010 [49] and which in print is tending to disappear, as the study revealed for the years 2017–2018 [40]. Nevertheless, on the Internet, its expansion is notable [51] and our research study demonstrates this increase in Twitter (*V* = 0.802).

The Internet has become a media of reference for interpersonal communication, the economy, education and entertainment [52], however, the social representation played out online is not neutral nor is it devoid of ideological components and power logics. Rather, segregationist discourse and behavior, especially towards women, continues to be reproduced under the veil of horizontality and equal opportunities [53,54]. Through analysis of coverage by the news media from the perspective of gender, this study shows that not only is female sport underrepresented, the media coverage carried out is biased and maximizes the presence of “gender appropriate” or “feminine· sports” [5,6,32,55].

In a subtle or not-so-subtle way and with an almost inoffensive appearance, these media practices continue to maintain the sports status quo of professional female athletes, for advertisers as well as for the audiences of sports competitions [56].

We cannot forget the positive role that social networks have, and especially Twitter, by opening up space for media coverage which amplifies that of traditional media [57]. Nevertheless, this study confirms that sports women are off the media radar and their presence in the new communication channels, as is the case with Twitter, is disproportionally lower than the monopolizing presence of men’s sports, and there is no reason to believe that this situation will change in the coming years [42].

## 5. Conclusions

Despite the fact that the international achievements and success of Spanish women athletes have grown exponentially in recent years and that the federation licenses of women have increased, sports women continue not being represented in the Spanish communication media. Media coverage of sports engaged in by women is lower as our study confirms, and the news panorama has not changed much in the last 35 years [28]. As a consequence, women are not in the sports media agenda, despite the fact that the new communication channels, such as Twitter, do not have space or time constraints. The results of this study shed light on how Twitter is at the service of the traditional media, reproducing the same gender bias and even increasing it.

This underrepresentation of female athletes in the media results in a lack of female role models in the digital media, which are used by the new generation as their main source of information. This in turn leads to perpetuating differential socialization in gender roles.

It has been confirmed that not all of the media behave in the same way, but in general, making women invisible and/or stereotyping them in the Twitter accounts of the sports media continues, going from female athletes to female non-athletes to continue maintaining a clear gender bias in sports news for their audience. The aforementioned confirms that sport media continues to be a highly masculine space that does not reflect the evolution that has indeed occurred in the practice of sport.

It has likewise confirmed that the coverage given is biased in maximizing the presence of “gender appropriate” sports, with the exception of football where women have notably increased their presence as athletes in recent years.

With the exception of football, in the rest of sports, sports achievements and successes do not imply a constant following in the Twitter accounts that are specialized in sports news. International Spanish women athletes who are standouts in sports such as badminton, weightlifting, karate or canoeing do not have the media impact that these sport accomplishments should have. The woman athlete continues to be at a clear disadvantage with respect to males because they still do not have sufficient stature to be featured in sports news, leading the Internet media to continue concealing the real situation of the female athletes.

## Figures and Tables

**Table 1 ijerph-17-05199-t001:** Sample distribution of gender according to communication media for tweets (n = 6544).

	Women		Men		Total
Media	%Column	%Row	%Column	%Row	%Column
*@Marca*	18.07%	2.01%	34.90%	97.99%	34.26%
*@MundoDeportivo*	59.84%	4.77%	47.24%	95.23%	47.72%
*@ElPaís_Deportes*	9.64%	4.45%	8.18%	95.55%	8.24%
*@ABC_Deportes*	12.45%	4.84%	9.67%	95.16%	9.78%
Total	100.00%	3.81%	100.00%	96.19%	100.00%

**Table 2 ijerph-17-05199-t002:** Cramer’s coefficient (V) of the variables “gender represented”, “sport”, “photo” and “media”.

	Gender Represented	Sport	Photo	Media
Gender Represented	-	0.403 *	0.069	0.068
Sport	0.403 *	-	0.046	0.140
Photo	0.069	0.046	-	0.312 *
Media	0.068	0.140	0.312 *	-

*V* > 0.3; * *V* > 0.6.

**Table 3 ijerph-17-05199-t003:** *p* values corresponding to the $\chi^{2}$ Tests of Independence of the variables “gender represented”, “sport”, “photo” and “media”.

	Gender Represented	Sport	Photo	Media
Gender Represented	-	0.000 **	0.000 **	0.000 **
Sport	0.000 **	-	0.015 *	0.000 **
Photo	0.000 **	0.015 *	-	0.000 **
Media	0.000 **	0.000 **	0.000 **	-

* *p* < 0.05; ** *p* < 0.00.

**Table 4 ijerph-17-05199-t004:** Sample distribution of gender according to the communication media and accompanying photo in tweets (n = 6544).

Media	Photo	Women		Men		Total
		%Column	%Row	%Column	%Row	%Column
*@Marca*	No	2.22%	0.72%	6.24%	99.28%	6.16%
	Yes	97.78%	2.09%	93.76%	97.91%	93.84%
	Total	100.00%	2.01%	100.00%	97.99%	100.00%
*@MundoDeportivo*	No	4.70%	0.80%	29.33%	99.20%	28.16%
	Yes	95.30%	6.33%	70.67%	93.67%	71.84%
	Total	100.00%	4.77%	100.00%	95.23%	100.00%
*@ElPaís_Deportes*	No	25.00%	2.82%	40.19%	97.18%	39.52%
	Yes	75.00%	5.52%	59.81%	94.48%	60.48%
	Total	100.00%	4.45%	100.00%	95.55%	100.00%
*@ABC_Deportes*	No	0.00%	0.00%	4.93%	100.00%	4.69%
	Yes	100.00%	5.08%	95.07%	94.92%	95.31%
	Total	100.00%	4.84%	100.00%	95.16%	100.00%
Total	No	5.62%	1.11%	19.80%	98.89%	19.26%
	Yes	94.38%	4.45%	80.20%	95.55%	80.74%
	Total	100.00%	3.81%	100.00%	96.19%	100.00%

**Table 5 ijerph-17-05199-t005:** Sample distribution of gender according to the sport mentioned in the tweets (n = 6544).

Sport	Women		Men		Total
	%Column	%Row	%Column	%Row	%Column
Football	20.88%	1.10%	74.14%	98.90%	72.11%
Basketball	5.62%	3.23%	6.67%	96.77%	6.63%
Motor racing	0.80%	0.93%	3.40%	99.07%	3.30%
Tennis	17.67%	22.00%	2.48%	78.00%	3.06%
Not reference to any sport	26.51%	33.17%	2.11%	66.83%	3.04%
Cycling	0.40%	0.51%	3.11%	99.49%	3.01%
Motorcycling	1.20%	1.53%	3.07%	98.47%	3.00%
Others	13.65%	22.97%	1.81%	77.03%	2.26%
Olympic movement	3.21%	16.00%	0.67%	84.00%	0.76%
Boxing	1.61%	8.00%	0.73%	92.00%	0.76%
Athletics	2.81%	15.56%	0.60%	84.44%	0.69%
Golf	0.00%	0.00%	0.67%	100.00%	0.64%
Handball	0.40%	2.94%	0.52%	97.06%	0.52%
Swimming	5.22%	92.86%	0.02%	7.14%	0.21%
Total	100.00%	3.81%	100.00%	96.19%	100.00%

**Table 6 ijerph-17-05199-t006:** Distribution of sample profiles of featured women (athletes or non-athletes) according to the communication media in the tweets (n = 249).

Media	Female Athletes		Non-Athletes		Total Women
	%Column	%Row	%Column	%Row	%Column
*@Marca*	18.12%	55.56%	18.02%	44.44%	18.07%
*@MundoDeportivo*	46.38%	42.95%	76.58%	57.05%	59.84%
*@ElPaís_Deportes*	17.39%	100.00%	0.00%	0.00%	9.64%
*@ABC_Deportes*	18.12%	80.65%	5.41%	19.35%	12.45%
Total	100.00%	55.42%	100.00%	44.58%	100.00%

**Table 7 ijerph-17-05199-t007:** Cramer’s Coefficient of the variables “profile of women represented” (athlete vs. non-athlete), “photo”, “sport” and “media.”

	Women’s Profile	Sport	Photo	Media
**Women’s Profile**	-	0.802 **	0.149	0.384 *****
**Sport**	0.802 **	-	0.254	0.295
**Photo**	0.149	0.254	-	0.284
**Media**	0.384 *	0.295	0.284	-

******V* > 0.3; ** *V* > 0.6

**Table 8 ijerph-17-05199-t008:** *p* values corresponding to $\chi^{2}$ Tests of Independence for the variables “profiles of women represented” (athlete or non-athlete), “photo” and “media.”

	Women’s Profile	Sport	Photo	Media
**Women’s Profile**	-	0.000 **	0.019 *	0.000 **
**Sport**	0.000 **	-	0.007 *	0.000 **
**Photo**	0.019 *	0.007 *	-	0.000 **
**Media**	0.000 **	0.000 **	0.000 **	-

* *p* < 0.05; ** *p* < 0.00.

**Table 9 ijerph-17-05199-t009:** Distribution of the profile of “women represented” (athletes vs non-athletes) according to the communication media and accompanying photo in tweet (n = 249).

Media	Photo	Female Athletes		Non-Athletes		Total Women
		%Column	%Row	%Column	%Row	%Column
*@Marca*	No	4.00%	100.00%	0.00%	0.00%	2.22%
	Yes	96.00%	54.55%	100.00%	45.45%	97.78%
	Total	100.00%	55.56%	100.00%	44.44%	100.00%
*@MundoDeportivo*	No	7.81%	71.43%	2.30%	28.57%	4.64%
	Yes	92.19%	40.97%	97.70%	59.03%	95.36%
	Total	100.00%	42.38%	100.00%	57.62%	100.00%
*@ElPaís_Deportes*	No	25.00%	100.00%	0.00%	0.00%	25.00%
	Yes	75.00%	100.00%	0.00%	0.00%	75.00%
	Total	100.00%	100.00%	0.00%	0.00%	100.00%
*@ABC_Deportes*	No	0.00%	0.00%	0.00%	0.00%	0.00%
	Yes	100.00%	80.65%	100.00%	19.35%	100.00%
	Total	100.00%	80.65%	100.00%	19.35%	100.00%
Total	No	8.70%	85.71%	1.77%	14.29%	5.58%
	Yes	91.30%	53.16%	98.23%	46.84%	94.42%
	Total	100.00%	54.98%	100.00%	45.02%	100.00%

**Table 10 ijerph-17-05199-t010:** Distribution of sample profiles of women represented (athlete vs non-athlete) according to the sport (n = 249).

Deporte	Female Athletes		Non-Athletes		Total Women
	%Column	%Row	%Column	%Row	%Column
Not reference to any sport	0.72%	1.49%	58.41%	98.51%	26.69%
Football	12.32%	32.69%	30.97%	67.31%	20.72%
Tennis	30.43%	95.45%	1.77%	4.55%	17.53%
Basketball	10.14%	100.00%	0.00%	0.00%	5.58%
Swimming	9.42%	100.00%	0.00%	0.00%	5.18%
Olympic movement	5.07%	87.50%	0.88%	12.50%	3.19%
Water polo	5.80%	100.00%	0.00%	0.00%	3.19%
Others	4.35%	85.71%	0.88%	14.29%	2.79%
Athletics	3.62%	71.43%	1.77%	28.57%	2.79%
Gymnastics	3.62%	100.00%	0.00%	0.00%	1.99%
Boxing	2.90%	100.00%	0.00%	0.00%	1.59%
Motorcycling	0.72%	33.33%	1.77%	66.67%	1.20%
Motor racing	0.00%	0.00%	1.77%	100.00%	0.80%
Ski	1.45%	100.00%	0.00%	0.00%	0.80%
Judo	1.45%	100.00%	0.00%	0.00%	0.80%
Cycling	0.00%	0.00%	0.88%	100.00%	0.40%
Handball	0.72%	100.00%	0.00%	0.00%	0.40%
Chess	0.72%	100.00%	0.00%	0.00%	0.40%
Canoeing	0.72%	100.00%	0.00%	0.00%	0.40%
Climbing	0.72%	100.00%	0.00%	0.00%	0.40%
Weightlifting	0.72%	100.00%	0.00%	0.00%	0.40%
Snow	0.72%	100.00%	0.00%	0.00%	0.40%
Surf	0.00%	0.00%	0.88%	100.00%	0.40%
Skating	0.72%	100.00%	0.00%	0.00%	0.40%
Volleyball	0.72%	100.00%	0.00%	0.00%	0.40%
Badminton	0.72%	100.00%	0.00%	0.00%	0.40%
Fencing	0.72%	100.00%	0.00%	0.00%	0.40%
Triathlon	0.72%	100.00%	0.00%	0.00%	0.40%
Total	100.00%	54.98%	100.00%	45.02%	100.00%

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
