# Peer review of "Gender Differences in Sports News Coverage on Twitter"

_ijerph, 2020, doi:10.3390/ijerph17145199_

Round 1

Reviewer 1 Report

I find the research very interesting. The role of women in sport is increasing every day in the world. Especially in Spain the results of women have increased notably. For that reason this paper is very opportune because as it is verified the informative pursuit of the sportswomen is not the appropriate one.

95-98 However, I am struck by the fact that the study was carried out between the months of March and July 2016. These are very old data. I thought that year was chosen to know the effect of the Olympic Games, but they were held in August. It would be interesting to extend the study and to know the effect of the Olympic Games.

In the discussion, reference is made to the 2016'2017 Women's First Division Football League.

The triumphs of the women's basketball team are mentioned, although in the years before the study they have very good results, the best results of the team were really from the Olympic Games of Rio de Janeiro (If we check the medal of the team we see the following data: Silver medal in the Olympic Games 2016; European Championship 2017 and 2019, Gold medal; World Championship 2014 and 2018 silver and bronze respectively).

It also refers that in the years 2018/19 women's sport was the leader in federal licenses.

If the study is carried out from March to July 2016, there is no reason to refer to this in the discussion.

So I recommend to focus on the period of the study or update it to current data. This option would be very interesting.

Author Response

Response to Reviewer 1 Comments

Point 1: 95-98 However, I am struck by the fact that the study was carried out between the months of March and July 2016. These are very old data. I thought that year was chosen to know the effect of the Olympic Games, but they were held in August. It would be interesting to extend the study and to know the effect of the Olympic Games 

Response 1: The period chosen, between the months of March and July 2016, precedes the Olympic Games (and does not include them in the analysis) for several reasons:

  • The routine period, March to July 2016, of information coverage refers to coverage that takes place during a normal season, without special events, except for the usual national and international leagues and tournaments. We decided to conduct an analysis on the media coverage of sports media during the routine period, because previous studies show that, during the Olympic Games, an extraordinary period of coverage, female athletes receive more and different media coverage than during any other sporting event. Therefore, it was necessary to analyse what such coverage is like during a "routine" period. That is, what is the day-to-day sports news coverage without a big sport and media event in the background?
  • The period before the Olympic Games is the key time for athletes to get sponsors. In order to get sponsors, media coverage is crucial.

(This change has been introduced in the procedure section p.3)

As the reviewer points out, it would be interesting to compare the 2016 pre-Olympic with the 2020 Olympic Games, but the suspension of the Olympic Games due to the health crisis in COVID19 has not allowed this longitudinal comparative analysis to be carried out.

Point 2: In the discussion, reference is made to the 2016/2017 Women's First Division Football League

Response 2: The reviewer refers to the fact that in the discussion he referred to the 2016-2017 Women's First Division Football League, while the data are earlier than 2016. Indeed, the change has been made and the accuracy of what was intended to be conveyed by that reference has been improved: Women's Football Club Association was established in 2015 to improve the visibility on television and social networks of women's football as a whole, rather than on an individual basis. This milestone meant that in August 2016 Iberdrola became the main sponsor of the championship and the competition began to have more visibility thanks to television coverage (according to LaLiga's own sources). (This change has been introduced in the discussion section p.8)

Subsequent references have been removed from the discussion so as not to draw erroneous conclusions. (This change has been introduced in the discussion section p.8)

Point 3: The triumphs of the women's basketball team are mentioned, although in the years before the study they have very good results, the best results of the team were really from the Olympic Games of Rio de Janeiro (If we check the medal of the team we see the following data: Silver medal in the Olympic Games 2016; European Championship 2017 and 2019, Gold medal; World Championship 2014 and 2018 silver and bronze respectively).

It also refers that in the years 2018/19 women's sport was the leader in federal licenses.

If the study is carried out from March to July 2016, there is no reason to refer to this in the discussion.

So I recommend to focus on the period of the study or update it to current data. This option would be very interesting.

Response 3: We have introduced the data for 2016. This change has been introduced in the discussion section p.8

Reviewer 2 Report

The paper studies the gender differences in the sport news coverage on Twitter. This is an interesting investigation and the research is timely. The data gathering, the study, and the analysis of the results are appropriate. The paper organization is clear. The language is good. The references include recent works. The discussion is relevant.

I only have the following remarks:

1) It is not clear how the data has been collected. Manually? Automatically?Please explain.

2) The data is from 2016. Four years are a significant period of time for such study. Isn't it possible to collect more recent data?

3) It would be good to compare these findings with similar findings in other countries.

Author Response

Please see the attachment or review the response to Reviewer 2 Comments here

Point 1: It is not clear how the data has been collected. Manually? Automatically? Please explain.

Response 1: The procedure for collecting the tweets has been done manually (this change has been introduced in the procedure section p.2-3)

Point 2:The data is from 2016. Four years are a significant period of time for such study. Isn't it possible to collect more recent data?

Response 2:The period chosen, between the months of March and July 2016, precedes the Olympic Games (and does not include them in the analysis) for several reasons:

  • The routine period, March to July 2016, of information coverage refers to coverage that takes place during a normal season, without special events, except for the usual national and international leagues and tournaments. We decided to conduct an analysis on the media coverage of sports media during the routine period, because previous studies show that, during the Olympic Games, an extraordinary period of coverage, female athletes receive more and different media coverage than during any other sporting event. Therefore, it was necessary to analyze what such coverage is like during a "routine" period. That is, what is the day-to-day sports news coverage without a big sport and media event in the background?
  • The period before the Olympic Games is the key time for athletes to get sponsors. In order to get sponsors, media coverage is crucial.

(This change has been introduced in the procedure section p.3)

As the reviewer points out, it would be interesting to compare the 2016 pre-Olympic with the 2020 Olympic Games, but the suspension of the Olympic Games due to the health crisis in COVID19 has not allowed this longitudinal comparative analysis to be carried out.

Point 3:It would be good to compare these findings with similar findings in other countries.

Response 3:Spain is one of the 10 countries in the world that uses Twitter the most and tweets the most. It is one of the countries that uses this social network the most. For this reason, we decided to analyze Twitter in particular, and the Spanish media, for this research.

So, given the magnitude of the tweets posted in the analyzed period, we only selected tweets published by each of the selected media, not the Retweets of Spanish media.

We decided that the best option was to analyze the tweets posted on Twitter by the four analyzed media during the four months of analysis.

To our knowledge, in the World, this is one of the first study that analyses the sports media coverage of female athletes on Twitter. In the USA, a study has already been published (Hull, 2017) about sports coverage with a gender perspective on Twitter, in a routine period, referred to in this research to identify potential similarities. In addition, in relation to the published studies with a gender perspective in traditional media, the most important studies have been analyzed and published in the USA. That is why most of the quotes are about U.S. studies as Cooky, Messner & Hextrum (2013) and Cooky (2018).

Reviewer 3 Report

I would like to start by highlighting the gender vision of the article. It is important to know the behaviors that happen on social media to see possible trends. Undoubtedly, women's sport has been gaining importance but it is still not at the height of men's in visibility, especially on television.

Regarding the article, I believe that the methodology and the results are right, well described and with a desire to be reproducible. I would like to know the reason why the contents related to "no reference to any sport" were not removed, because they are a high percentage. Perhaps you should have a study of what content is and how they fit in with the gender vision.

The depth of the interpretation gives me more doubts. I think that it would be necessary to give one more return to the data that profiles such as that of the Sports World offer, as well as trying to understand what type of photos they offer and trying to have a sample to know if there is a more aesthetic than descriptive vision of the protagonists.

Author Response

Please see the attachment or review the response to Reviewer 3 Comments

Point 1: I would like to know the reason why the contents related to "no reference to any sport" were not removed, because they are a high percentage. Perhaps you should have a study of what content is and how they fit in with the gender visión

Response 1: “No reference to any sport” is important here because this study confirm there is a large number of tweets about women non-athletes.“Athletes” refers to all the people who have to do with the world of sport. Non-athlete is those who have no relationship with the world of sport (e.g., singers, actors), although men practically do not appear in the sports media when they refer to men's sport. On the contrary, it is confirmed that there are a very large number of tweets that refer to women who would be included the group of people unrelated to the sports world, representing a prominent result of this study.

So the results confirm that media coverage continues to show a stereotypical and sexualised image of female athletes. But the results of our study go further by proposing that these biases shift towards non-athlete women, with whom female athletes share the media space of the specialized media, causing readers of these media continue to receive a stereotypical and biased image of "all" women, helping to maintain stereotypes about female athletes indirectly, and thus less obviously and more difficult to combat.  

The results of this study show a low objectification of female athletes in media coverage while it revealed how such objectification is maintained toward non-athlete women with whom female athletes share the informative space. It is also argued that these results cannot be considered as proof that female athletes are now safe from objectification.

Therefore, one of the conclusions presented in this study is that female athletes do share an informative space with non-athlete women (no reference to any sport), which is not the case for men, as it is mostly male athletes who are represented.

Point 2:The depth of the interpretation gives me more doubts. I think that it would be necessary to give one more return to the data that profiles such as that of the Sports World offer, as well as trying to understand what type of photos they offer and trying to have a sample to know if there is a more aesthetic than descriptive vision of the protagonists.

Response 2:In relation to the suggestion of content analysis of the results concerning the photographs, this would be an interesting line of future research that should be addressed with another methodology and an own analysis sheet constructed for that purpose. This suggestion will be taken into account for future research.
